# Functionality and Quality of Life with Parkinson’s Disease after Use of a Dynamic Upper Limb Orthosis: A Pilot Study

**DOI:** 10.3390/ijerph20064995

**Published:** 2023-03-12

**Authors:** María Jiménez-Barrios, Jerónimo González-Bernal, Esther Cubo, José María Gabriel-Galán, Beatriz García-López, Anna Berardi, Marco Tofani, Giovanni Galeoto, Martin J. A. Matthews, Mirian Santamaría-Peláez, Josefa González-Santos

**Affiliations:** 1Department of Health Sciences, University of Burgos, 09001 Burgos, Spain; 2Neurology Service, Burgos University Hospital, 09006 Burgos, Spain; 3Neurophysiology Service, Burgos University Hospital, 09006 Burgos, Spain; 4Department of Human Neurosciences, University of la Sapienza, 00188 Rome, Italy; 5Faculty of Health, School of Health Professions Peninsula Allied Health Centre, University of Plymouth, Derriford Rd., Plymouth PL6 8BH, UK

**Keywords:** Parkinson’s disease, dynamic elastomeric fabric orthosis, functionality, quality of life, non-pharmacological treatment

## Abstract

Parkinson’s disease (PD) is a chronic, neurodegenerative movement disorder, whose symptoms have a negative impact on quality of life and functionality. Although its main treatment is pharmacological, non-pharmacological aids such as the dynamic elastomeric fabric orthosis (DEFO) merit an evaluation. Our objective is to assess the DEFO in upper limb (UL) functional mobility and in the quality of life of PD patients. A total of 40 patients with PD participated in a randomized controlled crossover study, and were assigned to a control group (CG) and to an experimental group (EG). Both groups used the DEFO for two months, the experimental group the first two months of the study and the control group the last two. Motor variables were measured in the ON and OFF states at the baseline assessment and at two months. Differences from the baseline assessment were observed in some motor items of the Kinesia assessment, such as rest tremor, amplitude, rhythm or alternating movements in the ON and OFF states with and without orthosis. No differences were found in the unified Parkinson’s disease rating scale (UPDRS) or the PD quality-of-life questionnaire. The DEFO improves some motor aspects of the UL in PD patients but this does not translate to the amelioration of the standard of functional and quality-of-life scales.

## 1. Introduction

In the Global Burden of Diseases, Injuries and Risk Factors (GBD) study conducted in 2016, it was estimated that between the years 1990 and 2016 the number of people affected by Parkinson’s disease (PD) doubled worldwide, with an incidence rate of 8 to 18 people per 100,000 per year [1]. PD is defined as a chronic, neurodegenerative movement disorder, whose most characteristic motor symptoms are resting tremor, rigidity, and bradykinesia. Resting tremor is characterized by a prominent involuntary, rhythmic muscle movement in the distal upper limb (UL) at a frequency of about 4 to 6 Hz. Rigidity is an increased resistance to passive movement. The third most characteristic symptom is bradykinesia, characterized by slow movement and difficulty planning, initiating, and carrying out a movement [2]. Other non-motor symptoms such as sleep problems, constipation, anxiety, depression and fatigue may also appear [3]

The motor and non-motor symptoms of PD have negative repercussions on the quality of life and functionality of people with the disease. The burden of motor symptoms and impairment of some activities of daily living (ADLs), such as eating, hygiene and clothing, related to alterations in functional mobility, has been identified as one of the major predictors of quality of life with this disease [4,5,6].

Following the perspective of the International Classification of Functioning and Disability (ICF) of the World Health Organization (WHO), three interconnected levels of human functioning are differentiated: (1) Body functions and structures, physiological and psychological functions, and bodily and anatomical impairments; (2) Limitations in the performance of activities; and (3) Restrictions in participation in daily life [7,8]. The progression of PD leads to alterations in body function, limited performance of ADLs and increased dependence, while reducing quality of life [4,8,9,10].

As the disease progresses, the worsening of symptoms, such as tremor, rigidity and bradykinesia, leads to a deterioration of manual dexterity, which translates to a greater difficulty in performing some ADLs. The most commonly reported basic self-care activities affected by PD symptoms are bathing/showering, dressing, and grooming/personal hygiene. Other instrumental activities of daily living that are affected are driving, preparing food, shopping, and writing [11]. Therefore, the presence of these symptoms is closely related to a poorer quality of life [5,12].

The treatment of PD is mainly based on the administration of levodopa, whose efficacy decreases over time and can produce side effects such as motor fluctuations, dyskinesias and dopaminergic dysregulation syndrome. The onset of the disease and the variety of possible symptoms makes it difficult to design a therapeutic regimen for the treatment of the disease. So far, approved therapies have focused on compensatory approaches aimed at treating clinical symptoms. However, the current research is focused on delaying or halting disease progression and not only on temporary symptomatic relief. Currently, all therapies are directed toward ameliorating motor deficits by increasing dopamine, but unfortunately, this loses efficacy over time as dopaminergic neurodegeneration progresses, with symptoms worsening in the long-term. Therefore, new non-pharmacological therapies need to be assessed [13,14,15].

There are several non-pharmacological therapies design to reduce functional impairments of this disease and, although evidence of their efficacy is increasing, there is still a limited number of studies on them and on the necessary intervention doses [16,17]. New non-pharmacological therapies that can be easily implemented can complement pharmacological treatment in order to improve the patients’ functional mobility and quality of life. In this regard, the dynamic elastomeric fabric orthoses (DEFO, Figure 1) may be a suitable candidate for reducing motor symptoms and improving functional movement and quality of life in patients with PD. These types of devices were developed by dynamic movement orthoses^®^, led by clinical orthopedist and managing director Martin Matthews. They are custom-designed devices for the user’s limbs or other parts of his/her body. Through the application of traction forces, they bring the limb into a better biomechanical alignment, while allowing and guiding movement. The elastic fabric promotes the extension of fingers and wrist, the stability of the thumb and the supination or pronation of the forearm. In addition, due to the localized compression of the soft tissues and the stimulation of the dermal and proprioceptive receptors, it is possible to regulate motor activity, avoiding atrophy and muscle rigidity, improving the patient’s quality of life [18,19]. These orthoses, compared to other orthopedic devices, have demonstrated better tolerance and high user satisfaction [20,21,22].

This type of device has been effective in children with cerebral palsy (CP). In the study conducted by Pavão et al., the use, by children with CP, of a vest made of this material showed better postural stability when performing a manual reaching activity [23], and in another study, it showed improved balance, postural control, and manual dexterity [24]. On the other hand, wearing these devices on the foot and ankle improved balance and walking speed in multiple sclerosis [25,26], and pain and function in patients suffering with complex regional pain syndrome (CRPS) [16]. Stroke has been the condition in which the use of this UL orthosis has been most investigated, and several studies have shown positive effects on strength, manual dexterity, and UL functionality, which need to be confirmed in studies with larger sample sizes [27,28].

DEFOs have not yet been investigated in in a wide range of motor variables in PD. In the recent review, conducted by Son Nguyen, studies of different types of portable orthoses for UL tremor suppression were assessed, the majority being active orthoses (45%), followed by semi-active orthoses (35%), and passive orthoses (20%). All orthoses have proven to be effective in suppressing tremors, but several had inconveniencies such as being heavy and bulky, had not been evaluated in laboratory settings or were not yet commercially available [29].

Although current orthoses have proven to be effective in suppressing tremor, their clinical or home use is still limited. This limited their clinical or home use for suppressing tremor. Given these former results and lack of studies in PD, our main objective was to analyze the efficacy of a lighter device for the UL, such as the DEFO, in motor variables, functional mobility and quality of life in PD.

## 2. Materials and Methods

### 2.1. Participants

A longitudinal crossover study, with a control group and an experimental group, was carried out. Participants with PD were recruited by consecutive non-probability sampling from September to October 2021. The inclusion criteria were: male and female patients diagnosed with PD, who, during the recruitment period, were attending the Neurology Department of the Burgos University Hospital, in any of the stages of severity of the disease, who had tremor and rigidity as a consequence of the disease in at least one of the UL. Patients whose tremor was a consequence of another associated disease according to the neurologist’s judgment or/and those with scores less than or equal to 26 on the Montreal cognitive assessment (MoCA) were excluded [30].

The diagnosis of PD was established following the criteria established by the International Parkinson and Movement Disorder Society. The prerequisite for the application of these criteria is the presence of bradykinesia in combination with resting tremor, rigidity or both. In addition, at least two of the four supporting criteria had to be met: resting tremor, dramatic improvement with dopaminergic therapy, occurrence of dyskinesias as a consequence of levodopa or olfactory loss, or cardiac sympathetic denervation on myocardial scintigraphy [31,32].

Each participant signed a written informed consent approved by the Clinical Research Ethics Committee of the Health Area of Burgos and Soria (Spain) with reference number CEIM-2119/2019 before participating in the present study (ClinicalTrials.gov test number: NCT04815382). Likewise, the study was conducted in accordance with the ethical principles set forth in the Declaration of Helsinki [33].

### 2.2. Procedures

The calculation of the sample size was based on the tremor and rigidity improvement as the main variables of the study. Given alpha risk of 0.05 and a beta risk of 0.20, in bilateral contrast, it is estimated that 40 participants (20 for each group) were required to detect a minimum difference of 0.50 in the rigidity and tremor scores of the most affected UL using the unified Parkinson’s disease rating scale, motor subscale part III (UPDRS) [34]. Considering the 10% dropout rate during follow-up, a total sample of 40 patients was deemed necessary.

Using the Epidat 4.2 program, participants were randomly assigned to the experimental group (EG) or the control group (CG). The EG treatment protocol consisted of implementing the DEFO in the most affected UL for two months (intervention period), whereas subjects in the CG led life as usual during the first two months (control period). One month prior to the implementation of the DEFO, measurements of the size and posture of the UL were conducted for the customization of the orthosis in the participants of both groups. At the first visit, the sociodemographic and clinical data of the participants were collected, and their fulfilment of the inclusion criteria was ensured. The participants were instructed to maintain their prescribed dopaminergic medication regimen. The effects of the DEFO were evaluated during the ON state (under the benefit of levodopa) and during the OFF state (1 h before the next levodopa intake).

Motor assessments were conducted in the EG, at the end of the DEFO implementation period. Then, the DEFO was withdrawn and a second assessment was conducted two months later to evaluate if a carry-over effect was maintained during that time (Figure 2).

Several assessment tools were administered to evaluate the functional activity, quality of life, and manual dexterity of the subjects.

To obtain the primary outcomes, the unified Parkinson’s disease rating scale subscale II (UPDRS) was administered to assess functional activity consisting of 13 items. The score for each item is from 0 (normal) to 4 (worst), with a maximum score of 52 points, where higher scores indicate worse functional activity [35,36,37]. To assess the quality of life of each participant, the 39-item Parkinson’s disease questionnaire (PDQ-39) was administered, which consists of 29 items grouped into 8 domains: mobility, activities of daily living, emotional well-being, stigma, social support, cognition, communication, and grief and distress. Participants have to answer the questions based on their experience in the last four weeks. Each item is scored from 0 (never) to 4 (always). The maximum possible score is 156, with higher scores corresponding to worse quality of life [38,39].

For the assessment of UL dexterity, different motor aspects were evaluated. Subscale III of the UPDRS was administered, consisting of 17 items with a score range from 0 to 4 (from normal symptomatology to the most severe impairment), with a maximum score of 68 [35,36,37]. The Kinesia ONE motor assessment was used to collect and quantify the severity of motor symptoms such as tremor, bradykinesia, and dyskinesia. It provides an objective monitoring of Subscale III of the UPDRS. It is an electronic device consisting of software and a motion sensor. This sensor is positioned on the second finger of the hand during the time the patient performs a protocol of 12 tasks. The software scores each item from 0 (no symptoms) to 4 (severe impairment) [40].

Finally, the Purdue board test (PPT), the Minnesota manual dexterity test (MMDT) and the squares test (ST) were used to assess manual dexterity. The PPT consists of a two-column board that includes of 25 holes each, together with pins, washers, and rings located in four semicircles at the top of the board. The test is composed of four subtests that must be performed a total of three times, so that the total score is the average score obtained from the three attempts at each subtest. Thus, higher scores indicate better manual dexterity [41,42]. T, the abbreviated version of the MMDT, contains a rectangular wooden board that includes 60 holes distributed in 15 columns and 4 rows, as well as 60 circular pieces with one black and one red side of the same dimension as the holes in the board. It consists of two subtests that are performed a total of 4 times, obtaining as the final score, the average of the four attempts of each test. The final score is the time spent in performing the test, so the longer a patient spends, the worse the patient’s manual dexterity [43]. Finally, the ST contains a sheet of paper with four grids printed with 6 mm long squares. In the practice test, the patient must draw as many squares as possible for 10 s, while for the real test, he/she will have 30 s. The score is obtained for each hand by adding the number of dots drawn inside the squares, without touching the edges. Thus, a higher number of dots drawn indicates a better manual dexterity [44] (Figure 3).

## 3. Results

### 3.1. Baseline Characteristics of the Study Participants

The study had a simple crossover design, a total sample of 40 people with PD, 20 assigned to the CG, and 20 to the EG.

Table 1 summarizes the baseline socio-demographic characteristics of the participants according to the study group. Men represented 75% of the participants (*n* = 30), aged between 48 and 89 years, with a mean age of 71.00 ± 9.20 years and with 5.38 ± 4.23 years of disease evolution. The majority of participants (*n* = 35, 87.5%) lived accompanied at home, a minority lived alone at home (*n* = 4, 10%), and one in a religious community.

Of the participants, 62.5% (*n* = 25) had greater involvement in the right UL, while 37,5% (*n* = 15) had greater involvement in the left UL. Most participants (87.5%, *n* = 35) did not receive any type of non-pharmacological treatment and the rest, 12.5% (*n* = 5), attended physiotherapy, speech therapy, and/or occupational therapy.

Table 2 shows the Kinesia ONE^®^ measurements of action, resting tremor, and rigidity of the CG and EG participants before starting the intervention. The only differences between the groups are in resting tremor of the left UL in the OFF state.

### 3.2. Functionality and Quality of Life

Table 3 shows the differences observed in the baseline assessment with and without orthosis in the OFF state in the motor variables evaluated with Kinesia ONE^®^. Wearing the orthesis reduces “postural tremor” compared with not wearing it (*p* = 0.042), which can improve functionality and quality of life. The same effect may reduce the orthesis of “finger tapping amplitude” (*p* = 0.18) and of “speed in rapid alternating movements” with orthosis (*p* < 0.001).

Table 4 shows the differences in the motor variables evaluated with Kinesia ONE^®^ (Cleveland, OH, USA) in the baseline assessment with and without orthosis in the ON state. Wearing the orthosis reduces “resting tremor” compared with not wearing it (*p* = 0.009), which can improve functionality and quality of life. In the same way, the reduction with the orthosis of “finger tapping amplitude” (*p* = 0.027) and in the item “amplitude of rapid alternating movements” (*p* = 0.017) with orthosis can favor functionality and quality of life.

When comparing the change scores obtained on the UPDRS-II according to the patients’ condition and group type, no statistically significant differences were observed between the initial assessment and after two months of orthosis implementation in either the patients’ ON or OFF state. This means that no improvement in the UPDRS-II score was obtained after DEFO (Table 5).

Likewise, no differences (*p* = 0.933) were observed in the quality of life of the subjects after the implementation of the orthosis (Table 6).

## 4. Discussion

The aim of this study was to analyze the efficacy of the use of a DEFO for the UL on the functionality and quality of life of people with PD. The main findings of the present study are an immediate improvement after the implementation of the orthosis in the OFF and ON states of motor variables in the postural tremor task; only in the OFF state in the speed of rapid alternating movements and only in the ON state in the rhythm of hand movements and amplitude of rapid alternating movements. No differences were observed after two months of orthosis use in the improvement of functionality or in the quality of life of the patient with PD.

Neurological disorders, such as PD, are currently the leading source of disability in the world. The global burden of disease study estimated that the number of people with PD will double from about 7 million in 2015 to approximately 13 million in 2040. This estimation of the growth of the population with PD is worrying considering the amount of burden this disease carries for society [45].

The neurodegenerative effects of PD lead to a loss of functional mobility in balance, postural stability and gait, decreasing independence in the performance of activities, and compromising their participation both at home and in the community [6,46,47]. On the other hand, contextual factors such as age, the feeling of being a person with a disability, unemployment or perceived control are examples of personal and environmental factors that have a negative impact on the functional mobility and quality of life of the individual [6,7,8,47,48].

There have been many advances in the knowledge of the etiopathogenesis and in the symptomatic treatment of PD in recent years. However, there are no effective neuroprotective or disease-modifying therapies that slow disease progression and improve functionality and quality of life without producing side effects on the patient [5].

Due to the fact that pharmacological treatment loses its efficacy with the passage of time and produces side effects in the person and the lack of precise knowledge about the currently existing non-pharmacological therapies, it is necessary to implement new non-pharmacological therapies that allow an improvement in the functionality and quality of life of the patient [16,17].

All DEFOs are made in the same way, being able to be designed and adapted to the needs of the pathology and the user, so that they can be devices for UL, lower limbs or vest for the whole body. In diseases such as CP, different studies carried out with vests and meshes of these characteristics have demonstrated their efficacy on postural control, balance, walking speed, and manual dexterity [23,24,49]. In the study conducted by Yasukawa et al., in which DEFOs were implemented for UL in two cases with CP with hemiplegia and brachial plexus palsy, improved limb alignment and improved functionality of the affected UL were observed [18]. In the same way, they have also been effective in improving balance and walking speed in people with MS, as well as in improving pain and functionality of the lower limb in people with CRPS [20,25,26]. In a single case study conducted by Watson et al., the beneficial functional effects of a lycra orthosis in a multiple sclerosis patient were equivocal [50]. In another study of 16 patients with hemiparesis resulting from brain damage, the use of these devices showed a reduction in muscle tone and swelling, and improved wrist and finger movement [51]. Although some studies have shown that the use of these devices in people with brain damage improve strength, manual dexterity, and UL functionality, there is a need for studies with larger sample sizes [27,28]. These results coincide with those observed in the present study, in the sense that the implementation of the orthosis showed improvements on motor aspects of UL such as resting tremor, rhythm of hand movements or speed of rapid alternating movements, assessed with Kinesia, leading to an improvement in manual dexterity. No differences were observed in PDQ-39 and UPDRS-III scores on quality of life and functionality after orthosis implementation. Regarding the use of orthoses for tremor reduction, the review by Fromee et al. showed that the implementation of orthoses had a positive effect on the reduction in involuntary movement, being a complementary device to medical treatment. However, these orthoses turn out to be difficult to handle and unattractive, so they often lead to rejection by the patient. Therefore, there is a need to design orthosis that combines a tremor suppression mechanism with a soft, compact, and lightweight suppression system that increases patient acceptance [52]. Similarly in the review conducted by Mo et al., it was concluded that weight reduction in wearable orthosis for tremor reduction is an important research priority, as they have so far only been evaluated in patient cohorts or on the bench with simulated data and with very small samples, which may weaken the reliability of the data [53].

These findings should be considered within the context of their strengths and limitations; the results show new information about the efficacy of this type of orthosis in patients with PD. On the other hand, the evaluations have been carried out in the “ON” and “OFF” state of the disease, which gives us information on its effect in the different states of the disease. This device has proven to be a non-pharmacological treatment that is easy to implement, with high adherence to treatment and without any type of contraindication. With respect to the limitations, the nature of the intervention was such that the participants and investigators of the initial evaluation were not blinded, and it has not been possible to ascertain whether the results have been maintained in the long term due to the limited duration of the study.

## 5. Conclusions

The DEFO is a lightweight and easy-to-implement device. As a non-pharmacological treatment, it can be complementary to medication for the improvement of the motor aspects of UL in PD. Non-pharmacological interventions show promise in PD and need further studies.

## Figures and Tables

**Figure 1 ijerph-20-04995-f001:**
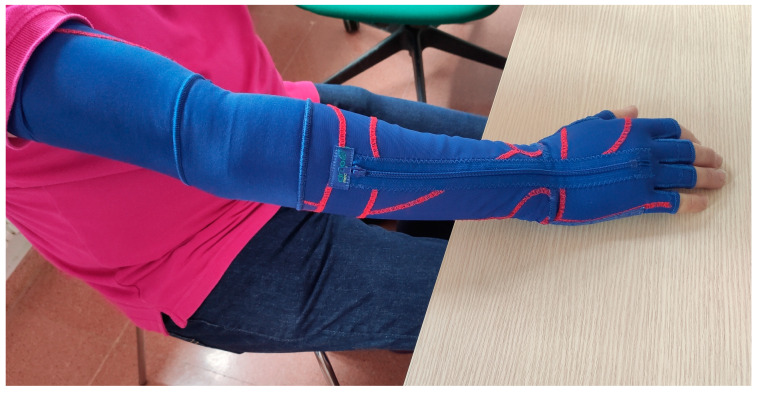
DEFO implemented in a patient with PD.

**Figure 2 ijerph-20-04995-f002:**
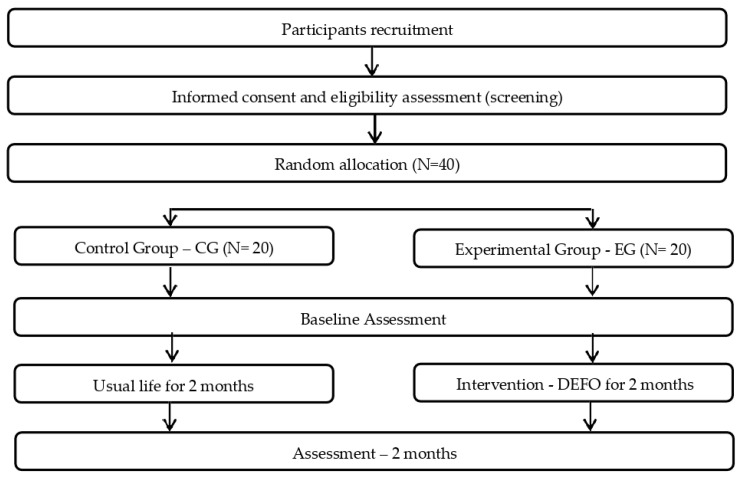
Study flow chart. DEFO: dynamic elastomeric fabric orthoses.

**Figure 3 ijerph-20-04995-f003:**
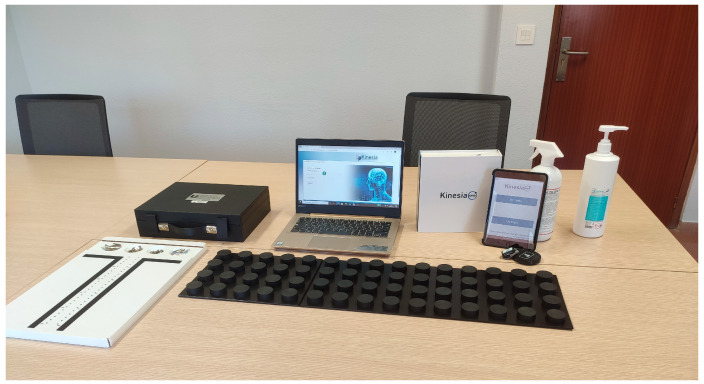
Evaluation kit (Purdue pegboard, Minnesota and Kinesia ONE).

**Table 1 ijerph-20-04995-t001:** Baseline characteristics of participants.

Variables	Total (*n* = 40)	CG (*n* = 20)	EG (*n* = 20)
**Age (years)**	71.00 ± 9.20	69.55 ± 12.31	72.18 ± 5.58
**Gender**			
Male	30	15	15
Female	10	3	7
**Most affected UL**			
Right	25	14	11
Left	15	3	12
**Years of disease evolution**	5.38 ± 4.23	4.72 ± 3.86	5.91 ± 4.52
**Current non-pharmacological treatment**			
Physiotherapy	2	0	2
Occupational therapy	0	0	0
Speech therapy	1	1	0
All	1	1	0
None	35	15	20
Others	1	1	0

Abbreviations: CG: control group; EG: experimental group; UL: upper limb.

**Table 2 ijerph-20-04995-t002:** Baseline UPDRS scores—resting and action tremor subtest.

UPDRS III	CG (*n* = 20)	EG (*n* = 20)	F	*p*-Value
**Action tremor ON**				
Right UL	0.625 ± 0.806	0.727 ± 702	0.173	0.680
Left UL	0.625 ± 0.619	0.863 ± 0.639	1.324	0.257
**Action tremor OFF**				
Right UL	0.875 ± 0.806	0.954 ± 0.843	0.085	0.772
Left UL	0.750 ± 0.577	1.181 ± 0.795	3.403	0.073
**Rest tremor OFF**				
Right UL	0.625 ± 0.619	1.136 ± 1.082	2.874	0.099
Left UL	0.500 ± 0.632	1.227 ± 0.922	7.391	**0.010**
**Rest tremor ON**				
Right UL	0.375 ± 0.619	0.818 ± 0.906	2.845	0.100
Left UL	0.375 ± 0.500	0.772 ± 0.812	2.995	0.092

CG: control group; EG: experimental group; *p*-value < 0.05.

**Table 3 ijerph-20-04995-t003:** Kinessia OFF state, with and without orthoses—baseline assessment (*n* = 40).

Variables		Mean	SD	*p*-Value
Rest tremor	Without orthoses	1.102	0.926	0.378
	With orthoses	0.956	0.900	
Postural tremor	Without orthoses	0.864	0.761	**0.042**
	With orthoses	0.621	0.631	
Kinetic tremor	Without orthoses	1.150	0.506	0.934
	With orthoses	1.136	0.952	
Finger tapping—speed	Without orthoses	1.888	1.012	0.834
	With orthoses	1.869	1.010	
Finger tapping—amplitude	Without orthoses	2.302	0.921	**0.018**
	With orthoses	2.016	0.988	
Finger tapping—rhythm	Without orthoses	1.352	1.041	0.719
	With orthoses	1.302	0.983	
Hand movements—speed	Without orthoses	2.030	0.767	0.300
	With orthoses	2.119	0.699	
Hand movements—amplitude	Without orthoses	1.400	0.767	0.948
	With orthoses	1.408	0.926	
Hand movements—rhytm	Without orthoses	0.891	0.651	0.669
	With orthoses	0.850	0.683	
Alternating quick movements—speed	Without orthoses	2.348	0.774	**<0.001**
	With orthoses	1.317	0.744	
Alternating quick movements–amplitude	Without orthoses	1.251	0.702	0.307
	With orthoses	1.317	0.744	
Alternating quick movements—rhytm	Without orthoses	1.285	1.247	0.367
	With orthoses	1.117	1.005	

Paired samples *t*-test; *p*-value < 0.05. SD: standard deviation.

**Table 4 ijerph-20-04995-t004:** Kinessia ON state, with and without orthoses—baseline assessment (*n* = 40).

Variables		Mean	SD	*p*-Value
Rest tremor	Without orthoses	0.997	0.830	**0.009**
	With orthoses	0.652	0.716	
Postural tremor	Without orthoses	0.832	0.877	0.432
	With orthoses	0.744	0.619	
Kinetic tremor	Without orthoses	1.184	0.421	0.416
	With orthoses	1.128	0.426	
Finger tapping—speed	Without orthoses	1.910	0.991	0.806
	With orthoses	1.878	1.018	
Finger tapping—Amplitude	Without orthoses	2.273	0.932	**0.027**
	With orthoses	1.952	1.051	
Finger tapping—rhythm	Without orthoses	1.171	0.963	0.319
	With orthoses	0.997	0.795	
Hand movements—speed	Without orthoses	2.089	0.743	0.922
	With orthoses	2.097	0.672	
Hand movements—Amplitude	Without orthoses	1.605	0.865	0.685
	With orthoses	1.542	0.899	
Hand movements—rhythm	Without orthoses	1.021	0.798	**0.051**
	With orthoses	0.085	0.589	
Alternating quick movements—speed	Without orthoses	2.278	0.720	0.198
	With orthoses	2.984	3.219	
Alternating quick movements—amplitude	Without orthoses	1.084	0.831	**0.017**
	With orthoses	1.265	0.696	
Alternating quick movements—rhythm	Without orthoses	1.294	1.319	0.479
	With orthoses	1.157	1.036	

Paired samples *t*-test; *p*-value < 0.05. SD: standard deviation.

**Table 5 ijerph-20-04995-t005:** Inter-group comparison of UPDRS-II differential score according to the type of group using ANCOVA.

Variables	Group	Mean	SD	MS	F	*p*-Value	ɳ^2^
OFF—with orthoses	CG	0.000	0.000	40.267	1.629	0.208	0.033
	EG	−1.914	5.907				
OFF—without orthoses	CG	0.500	7.033	87.233	2.546	0.117	0.048
	EG	−2.263	5.217				
ON—with orthoses	CG	0.000	0.000	31.762	1.430	0.238	0.029
	EG	−10.742	5.606				
ON—without orthoses	CG	0.187	5.344	7.123	0.290	0.593	0.006
	EG	−0.631	4.732				

CG: control group (*n* = 20); EG: experimental group (*n* = 20); SD: standard deviation, *p*-value < 0.05.

**Table 6 ijerph-20-04995-t006:** Inter-group comparison of PDQ-39 differential score between pre-test and post-test.

PDQ-39	*n*	Mean	SD	MS	F	*p*-Value	ɳ^2^
CG	16	−0.625	6.830	0.441	0.007	0.933	0.000
EG	22	−0.818	8.313				

CG control group (*n* = 20), EG: experimental group (*n* = 20), SD: standard deviation, *p*-value < 0.05.

## Data Availability

Not applicable.

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
