# Peer review of "Functionality and Quality of Life with Parkinson’s Disease after Use of a Dynamic Upper Limb Orthosis: A Pilot Study"

_ijerph, 2023, doi:10.3390/ijerph20064995_

Round 1

Reviewer 1 Report

113 Why a crossover study?. A parallel design is better because of the carryover effects. The only advantage of a crossover design is that a smaller sample size is needed.

117 recruited PD patients in any stages of the disease of the disease. Are there no exclusion criteria.

120 Appart from tremor for another pathology, what were the other exclusion criteria?. ,Age, life expectancy, cognitive impairment, functional impairment?

200 N 40. How was calculated the sample size

274 The main findings of the present study showed an immediate improvement after the implementation of the orthosis, in OFF and ON state in symptoms such as postural tremor and finger tapping amplitude; only in OFF state in the speed of rapid alternating movements and only in ON state, in the rhythm of hand movements and amplitude of rapid alternating movements.  What's the basis of this assertion?. According to the results and the figures displayed in the Tables it is difficult to agree with this statement

Too much written in the discussion section with little reference to this trial.

Author Response

University of Burgos, Paseo Comendadores s/n.

Burgos, 09001, Spain

Tel. (+34) 947499108

Email: mariajb@ubu.es

Subject: Submissions Needing Revision

Dear editor.

Thank you very much for inviting us to submit a revised version of our manuscript (ijerph-2078258) entitled: “Functionality and quality of life in Parkinson´s Disease after use of a dynamic upper limb orthosis”.

We have checked our manuscript according to the Academic Editor, the reviewers’ comments and the Journal requirements. We have also responded comments from reviewers point by point.

We would be very grateful if you could consider our manuscript to be published in your journal.

  • We have included a revised manuscript file.

Yours sincerely,

María Jiménez Barrios

Response to Reviewer 1

First of all, we would like to express our sincere gratitude for all comments and suggestions received from the Reviewer 1. This information has certainly enriched the text for its best understanding, thank you very much indeed. We have clarified the reviewer’s questions. We have introduced the required changes both in our answers to the specific comments and in the final manuscript.

113 Why a crossover study?. A parallel design is better because of the carryover effects. The only advantage of a crossover design is that a smaller sample size is needed.

Response: Thank you very much for pointing it out.

Because we had a small sample of 40 people with the established inclusion criteria, it was decided to carry out a crossover study to double the number of participants in the study and obtain statistically significant results.

117 recruited PD patients in any stages of the disease of the disease. Are there no exclusion criteria.

Response: Thank you very much for pointing it out. We have clarified this aspect in the participants section.

“Patients whose tremor was a consequence of another associated disease according to the neurologist's judgment, or/and with scores less or equal than 26 in the Montreal Cognitive Assessment (MoCA) were excluded”

120 Appart from tremor for another pathology, what were the other exclusion criteria?. ,Age, life expectancy, cognitive impairment, functional impairment?

Thank you very much for pointing it out. We have clarified this aspect in the participants section.

“With scores less or equal than 26 in the Montreal Cognitive Assessment (MoCA) were excluded”.

On the same day the measurements were taken, a screening was previously carried out with the MoCA.

200 N 40. How was calculated the sample size

Thank you very much for pointing it out. We have clarified this aspect in the procedures section.

The calculation of the sample size will be based on the tremor and rigidity improvement, as main variables of the study. Given alpha risk of 0.05 and a beta risk of 0.20, in bilateral contrast, it is estimated that 40 participants (20 for each group) will be required to detect a mínimum difference of 0.50 in rigidity and tremor of the most affected UL using the Unified Parkinson's Disease Rating Scale, motor subscale part III (UPDRS) (30). Considering a 10% dropout rate during follow-up, a total sample of 44 patients will be included”

274 The main findings of the present study showed an immediate improvement after the implementation of the orthosis, in OFF and ON state in symptoms such as postural tremor and finger tapping amplitude; only in OFF state in the speed of rapid alternating movements and only in ON state, in the rhythm of hand movements and amplitude of rapid alternating movements.  What's the basis of this assertion?. According to the results and the figures displayed in the Tables it is difficult to agree with this statement

Too much written in the discussion section with little reference to this trial.

Thank you very much for pointing it out. This aspect can be verified in tables 5 and 6 of the manuscript

When comparing the change scores obtained on the UPDRS-II according to the patient's condition and group type, no statistically significant differences were observed between the initial assessment and after two months of orthosis implementation in either the patient's ON or OFF state.

Likewise, no differences (p=.933) were observed in the quality of life of the subjects after the implementation of the orthosis.

Reviewer 2 Report

I have studied the research in detail. I thank the authors for their efforts.

The introduction is too short for readers. This section should be improved especially with regard to the rational between DEFO, symptom and quality of life. I propose to expand the information on tremor, rigidity and bradykinesia; more directional literature in L.55  “As the disease progresses, the worsening of symptoms such as tremor, rigidity and bradykinesia, leads to a deterioration of manual dexterity that translates into greater difficulty in performing some ADLs” ; which ADLs are most affected?

Please add exclusion criteria in the materials and methods

Add a detailed description of the research equipment used (manufacturer, country, number, symbol, etc.).

Add images for instrument used (Kinesia ONE motor assessment) and dynamic elastomeric fabric orthosis

Add a description of the applied statistical procedures (separate subchapter). For example: why did you choose the t-test? why not the anova? what software did you use?

Discussions should be expanded with more references to the literature. And I highly recommend add the strengthes and limitations in this study at the end of the discussion section.

Authors should add a limitations section. The small number of the sample analyzed does not support the conclusions. should be considered as a pilot study and accordingly change the title to “Functionality and quality of life in Parkinson´s Disease after use of a dynamic upper limb orthosis: a pilot study”

Author Response

University of Burgos, Paseo Comendadores s/n.

Burgos, 09001, Spain

Tel. (+34) 947499108

Email: mariajb@ubu.es

Subject: Submissions Needing Revision

Dear editor.

Thank you very much for inviting us to submit a revised version of our manuscript (ijerph-2078258) entitled: “Functionality and quality of life in Parkinson´s Disease after use of a dynamic upper limb orthosis”.

We have checked our manuscript according to the Academic Editor, the reviewers’ comments and the Journal requirements. We have also responded comments from reviewers point by point.

We would be very grateful if you could consider our manuscript to be published in your journal.

  • We have included a revised manuscript file.

Yours sincerely,

María Jiménez Barrios

Response to Reviewer 2

First of all, we would like to express our sincere gratitude for all comments and suggestions received from the Reviewer 2. This information has certainly enriched the text for its best understanding, thank you very much indeed. We have clarified the reviewer’s questions. We have introduced the required changes both in our answers to the specific comments and in the final manuscript.

I have studied the research in detail. I thank the authors for their efforts.

The introduction is too short for readers. This section should be improved especially with regard to the rational between DEFO, symptom and quality of life. I propose to expand the information on tremor, rigidity and bradykinesia; more directional literature in L.55  “As the disease progresses, the worsening of symptoms such as tremor, rigidity and bradykinesia, leads to a deterioration of manual dexterity that translates into greater difficulty in performing some ADLs” ; which ADLs are most affected?

Response: Thank you very much for pointing it out. We have clarified this aspect in the introduction section.

“The elastic fabric promotes the extension of fingers and wrist, the stability of the thumb and the supination or pronation of the forearm. In addition, due to the localized compression of the soft tissues and the stimulation of the dermal and proprioceptive receptors, it is possible to regulate motor activity, avoiding atrophy and muscle rigidity, while improving the patient's quality of life”.

“Resting tremor is characterized by a prominent involuntary, rhythmic muscle movement in the distal upper limb (UL) at a frequency of about 4 to 6 Hz. Rigidity is an increased resistance to passive movement. The third most characteristic symptom is bradykinesia, characterized by slow movement and difficulty planning, initiating, and carrying out a movement”

“The most commonly reported basic self‐care activities affected by PD symptoms are bathing/showering, dressing and grooming/personal hygiene. Other instrumental activities of daily living affected are driving, preparing food, shopping, and writing.”

Please add exclusion criteria in the materials and methods

Response: Thank you very much for pointing it out. We have clarified this aspect in materials and methods section.

“Patients whose tremor was a consequence of another associated disease according to the neurologist's judgment, or/and with scores less or equal than 26 in the Montreal Cognitive Assessment (MoCA) were excluded”.

Add a detailed description of the research equipment used (manufacturer, country, number, symbol, etc.).

Response: Thank you very much for pointing it out. We have clarified this aspect in the introduction section.

“These types of devices were developed by Dynamic Movement Orthoses®, led by clinical orthopedist and managing director Martin Matthews”.

Add images for instrument used (Kinesia ONE motor assessment) and dynamic elastomeric fabric orthosis

Response: Thank you very much for pointing it out. We have added two figures about of the DEFO and the evaluation kit in introduction and procedures sections.

Add a description of the applied statistical procedures (separate subchapter). For example: why did you choose the t-test? why not the anova? what software did you use?

Data are presented as number of cases (% of total) and as mean ± standard deviation (SD). Statistical analysis was performed with the software SPSS version 25 (IBM-Inc., Chicago, IL, USA). Statistical significance was determined with a p-value < 0.05.

The differences in the total score of Kinesia, quality of life and UPDRS II, between groups, were contrasted by means of bivariate analysis, performing a t test for independent samples, having as a fixed factor the group to which each user belonged, and an analysis test. of covariance, of the posttest-pretest differential score, for the longitudinal comparison of both group

Discussions should be expanded with more references to the literature. And I highly recommend add the strengths and limitations in this study at the end of the discussion section.

Response: Thank you very much for pointing it out. We have clarified this aspect in the discussion section.

“These findings should be considered within the context of their strengths and limitations; the results show new information about the efficacy of this type of orthosis in patients with PD. On the other hand, the evaluations have been carried out in the “ON” and “OFF” state of the disease, which gives us information on its effect in the different states of the disease.This device has proven to be a non-pharmacological treatment that is easy to implement, with high adherence to treatment and without any type of contraindication. With respect to the limitations, the nature of the intervention the participants and investigators of the initial evaluation were not blinded and it has not been possible to ascertain whether the results have been maintained in the long term due to the limited duration of the study”

Authors should add a limitations section. The small number of the sample analyzed does not support the conclusions. should be considered as a pilot study and accordingly change the title to “Functionality and quality of life in Parkinson´s Disease after use of a dynamic upper limb orthosis: a pilot stud

Response: Thank you very much for pointing it out. We have changed the tittle.

Reviewer 3 Report

Thank you for Maria Jiménez-Barrios et al submitted us a paper with an functionality and quality of life in Parkinson´s Disease after use of a dynamic upper limb orthosis. The authors indicate dynamic elastomeric fabric orthosis (DEFO) can be considered an easy-to-implement, lightweight and easy-to-implement device that can be complemented as a non-pharmacological treatment to medication for the improvement of the motor aspects of Upper Limb (UL) in Parkinson´s Disease.

After reading the manuscript, it was originally decided that reconsider after major revision, but the supplementary materials were lacking, and the manuscript was finally considered to be rejected.

Maria Jiménez-Barrios et al submitted us a paper with an functionality and quality of life in Parkinson´s Disease after use of a dynamic upper limb orthosis. The authors indicate dynamic elastomeric fabric orthosis (DEFO) can be considered an easy-to-implement, lightweight and easy-to-implement device that can be complemented as a non-pharmacological treatment to medication for the improvement of the motor aspects of Upper Limb (UL) in Parkinson´s Disease. After reviewing the manuscript, we original opinion was “reconsider after major revision”. However, it was found that there was a lack of supplementary materials, and the final opinion was to reject the manuscript. The reasons for rejecting the manuscript raise as follows:

1. The abbreviation and complete spelling of some words are not consistent in this paper, which makes the content look confused. When the word appears in the article for the first time, it should be spelled completely, and abbreviations should be introduced, then abbreviations should be used instead in the following pages. For instance:

(1) the “ADLs”, “SG”, ”CG”, EG”and“DEFO”.

(2) Tables 5 and 6: Misspelling of abbreviations in “Group” and “Notes” sections.

2. Maybe you could describe in detail the“some kinesia items” in the “ABSTRACT” part.

3. Lines 1-2 of paragraph 5 on page 3, description about “Using the Epidat 4.2 program, participants were randomly assigned to the CG or the CG.”Are there errors in spelling of groupings?

4. Why was the sample size shown in “Figure 1” 44 participants, while only 40 participants were included in“Results”?

5. “First evaluation”,“Baseline assessment”and“Basal assesment” appear disorganized with multiple occurrences in the text. The “first evaluation” was defined as the assessment at baseline or 2 months after enrollment?

6. The following points exist for Table 1:

(1) Age” without units.

(2) In “Table 1”, 15 participants are shown had greater involvement in the left UL. However, lines 1-2 of paragraph 4 on page 5, description about“while 35% (n=13) had greater involvement in the left UL”. Why are the values inconsistent?

7. Lines 3-7 of paragraph 1 on page 9, the sentence was too long to be understood.

8. Lack of supplementary materials related to the tables.

Author Response

Thank you very much for inviting us to submit a revised version of our manuscript (ijerph-2078258) entitled: “Functionality and quality of life in Parkinson´s Disease after use of a dynamic upper limb orthosis”.

We have checked our manuscript according to the Academic Editor, the reviewers’ comments and the Journal requirements. We have also responded comments from reviewers point by point.

We would be very grateful if you could consider our manuscript to be published in your journal.

  • We have included a revised manuscript file.

Yours sincerely,

María Jiménez Barrios

Response to Reviewer 3

First of all, we would like to express our sincere gratitude for all comments and suggestions received from the Reviewer 3. This information has certainly enriched the text for its best understanding, thank you very much indeed. We have clarified the reviewer’s questions. We have introduced the required changes both in our answers to the specific comments and in the final manuscript.

  1. The abbreviation and complete spelling of some words are not consistent in this paper, which makes the content look confused. When the word appears in the article for the first time, it should be spelled completely, and abbreviations should be introduced, then abbreviations should be used instead in the following pages. For instance:

(1) the “ADLs”, “SG”, ”CG”, EG”and“DEFO”.

(2) Tables 5 and 6: Misspelling of abbreviations in “Group” and “Notes” sections.

Response: Thank you very much for pointing it out. We have clarified these aspects throughout the text and have made the necessary changes in tables 5 and 6.

  1. Maybe you could describe in detail the “some kinesia items”in the “ABSTRACT” part.

Response: Thank you very much for pointing it out. We have clarified this aspect in the abstract section.

“Differences from the baseline assesment were observed in some motor items of the Kinesia assessment like rest tremor, amplitude, rhythm or alternating quick movements in ON and OFF state with and without orthosis in the baseline assessment”

  1. Lines 1-2 of paragraph 5 on page 3, description about “Using the Epidat 4.2 program, participants were randomly assigned to the CGor the CG.”Are there errors in spelling of groupings?

Response: Thank you very much for pointing it out. We have clarified this aspect in those lines.

“Using the Epidat 4.2 program, participants were randomly assigned to the EG or the CG”

  1. Why was the sample size shown in “Figure 1”44 participants, while only 40 participants were included in“Results”?

Response: Thank you very much for pointing it out. We have clarified this aspect in figure 1.

  1. “First evaluation”,“Baseline assessment”and“Basal assesment”appear disorganized with multiple occurrences in the text. The “first evaluation” was defined as the assessment at baseline or 2 months after enrollment?

Thank you very much for pointing it out. We have clarified these aspects throughout the text.

The correct term is Baseline Assessment, defined as the first time data collection is performed in patients.

  1. The following points exist for Table 1:

(1) “Age” without units.

(2) In “Table 1”, 15 participants are shown had greater involvement in the left UL. However, lines 1-2 of paragraph 4 on page 5, description about“ while 35% (n=13) had greater involvement in the left UL”. Why are the values inconsistent?

Thank you very much for pointing it out. We have clarified these aspects in table 1 and in the text.

  • “Age (years)”
  • Of the participants, 62.5% (n = 25) had greater involvement in the right UL, while 37,5% (n = 15)
  1. Lines 3-7 of paragraph 1 on page 9, the sentence was too long to be understood.

Thank you very much for pointing it out. We have clarified these lines.

“Table 4 shows the differences in the motor variables evaluated with Kinesia ONE®.. in the baseline assessment with and without orthosis in the ON state. Wearing the orthesis reduces “resting tremor” compared with not wearing it (p = .009) which can improve functionality and quality of life. In the same way, the reduction with the orthesis of “finger tapping amplitude” (p= .027) and in the item “amplitude of rapid alternating movements” (p = .017) with orthosis can favor functionality and quality of life”

  1. Lack of supplementary materialsrelated to the tables.

Thank you very much for pointing it out. We have added images and clarified some terms of the tables

Reviewer 4 Report

The research presented in the article certainly belongs to the Journal important topics. Authors research contributes to the problematics of DEFO treatment efficiency. They proved an immediate improvement after the implementation of the orthosis, in OFF and ON states in symptoms such as postural tremor and finger tapping amplitude,  in OFF state in the speed of rapid alternating movements, and in ON state, in the rhythm of hand movements and amplitude of rapid alternating movements. This research may help with the faster implementation of new non-pharmacological therapies that allow an improvement in the functionality and quality of  life of the patient. It may decrease needs for intensive drugs treatment.

The article has a good structure, methodology is well described, experiments are statistically evaluated. Authors used 48 references.

However, there are several formal and grammatical errors that need to be corrected :

line 53 and 57: ADLs abbreviation is used without explanation of its meaning,

line 68: Therefore, A complementary...  correctly shall be ... a complementary...

line 113: The following sentence shall be corrected, a verb is missing:  Longitudinal crossover study, with a control group and an experimental group.

Please, correct proper use of CG, and EG.  SG and GE seem to be wrong words. It is used many times as it follows

line 137-138: .. to the CG or the CG. 

line 202: ,,.18 belonging to the CG and 22 to the GE. 

GE instead of EG occurs in those lines: 202, 216, in tables 5, 6

line 146-147:  correct that sentence as follows ...during the ON state (under the benefit of levodopa) and during the wearing OFF state (1 hour before next levodopa intake).    I am also not sure that wearing is a good word to use in that sentence.

Author Response

Response to Reviewer 4

First of all, we would like to express our sincere gratitude for all comments and suggestions received from the Reviewer 4. This information has certainly enriched the text for its best understanding, thank you very much indeed. We have clarified the reviewer’s questions. We have introduced the required changes both in our answers to the specific comments and in the final manuscript.

line 53 and 57: ADLs abbreviation is used without explanation of its meaning,

Response: Thank you very much for pointing it out. We have clarified this aspect.”

“The burden of motor symptoms and impairment of some activities of daily living (ADLs) as eating, hygiene and clothing related to alterations in functional mobility has been identified as one of the major predictors of quality of life in this disease.”

line 68: Therefore, A complementary...  correctly shall be ... a complementary...

Response: Thank you very much for pointing it out. We have clarified this aspect

“Therefore, a complementary alternative…”

line 113: The following sentence shall be corrected, a verb is missing:  Longitudinal crossover study, with a control group and an experimental group.

Response: Thank you very much for pointing it out. We have clarified this aspect

“A longitudinal crossover study, with a control group and an experimental group was carried out.”

Please, correct proper use of CG, and EG.  SG and GE seem to be wrong words. It is used many times as it follows

line 137-138: .. to the CG or the CG. 

line 202: ,,.18 belonging to the CG and 22 to the GE. 

GE instead of EG occurs in those lines: 202, 216, in tables 5, 6

Response: Thank you very much for pointing it out. We have changed these mistakes in the text.

line 146-147:  correct that sentence as follows ...during the ON state (under the benefit of levodopa) and during the wearing OFF state (1 hour before next levodopa intake).    I am also not sure that wearing is a good word to use in that sentence.

Response: Thank you very much for pointing it out. We have changed this sentence.

“The effectiveness of DEFO was evaluated during the ON state (under the benefit of levodopa) and during the wearing OFF state (1 hour before next levodopa intake)”

Round 2

Reviewer 2 Report

Accetable for publication if it appropriate for the editor and other reviewers

Author Response

The appreciations indicated throughout the study have allowed me to improve it. 
Thank you very much

Reviewer 3 Report

Dear Authors,

Thank you for submitted us a paper with an functionality and quality of life in Parkinson´s Disease after use of a dynamic upper limb orthosis. The authors indicate dynamic elastomeric fabric orthosis (DEFO) can be considered an easy-to-implement, lightweight and easy-to-implement device that can be complemented as a non-pharmacological treatment to medication for the improvement of the motor aspects of Upper Limb (UL) in Parkinson´s Disease. After reviewing the manuscript, we opinion was “ Accept after minor revision”. The comment on the revision of the manuscript is as follow:

1. Table 5 “GC: Control group (n=20)” 

  The abbreviation of "Control group" is incorrect. It should be "CG".

Kind regards,

Nian xiong

Author Response

Thank you very much for pointing it out. We have clarified this aspect in table 5.

"CG: Control group"